# The Depression Anxiety Stress Scale 21: Development and Validation of the Depression Anxiety Stress Scale 8-Item in Psychiatric Patients and the General Public for Easier Mental Health Measurement in a Post COVID-19 World

**DOI:** 10.3390/ijerph181910142

**Published:** 2021-09-27

**Authors:** Amira Mohammed Ali, Abdulmajeed A. Alkhamees, Hiroaki Hori, Yoshiharu Kim, Hiroshi Kunugi

**Affiliations:** 1Department of Behavioral Medicine, National Institute of Mental Health, National Center of Neurology and Psychiatry, Tokyo 187-8553, Japan; hori@ncnp.go.jp (H.H.); kim@ncnp.go.jp (Y.K.); 2Department of Psychiatric Nursing and Mental Health, Faculty of Nursing, Alexandria University, Alexandria 21527, Egypt; 3Department of Medicine, College of Medicine and Medical Sciences, Qassim University, Buraydah 52571, Al Qassim, Saudi Arabia; A.alkhamees@qu.edu.sa; 4Department of Psychiatry, Teikyo University School of Medicine, Tokyo 173-8605, Japan; hkunugi@med.teikyo-u.ac.jp; 5Department of Mental Disorder Research, National Institute of Neuroscience, National Center of Neurology and Psychiatry, Tokyo 187-8502, Japan

**Keywords:** Coronavirus disease 2019/COVID-19, Depression Anxiety Stress Scales-21/DASS-21, DASS-8, shortened version*, shorter version* of the DASS-21, psychiatric disorders, factorial structure/psychometric properties/structural validity/validation, ceiling effect, measurement invariance/multigroup analysis, psychological distress, discriminant validity, item coverage, good predictive validity, Arabic, Arab, Saudi Arabia

## Abstract

Despite extensive investigations of the Depression Anxiety Stress Scales-21 (DASS-21) since its development in 1995, its factor structure and other psychometric properties still need to be firmly established, with several calls for revising its item structure. Employing confirmatory factor analysis (CFA), this study examined the factor structure of the DASS-21 and five shortened versions of the DASS-21 among psychiatric patients (N = 168) and the general public (N = 992) during the COVID-19 confinement period in Saudi Arabia. Multigroup CFA, Mann Whitney W test, Spearman’s correlation, and coefficient alpha were used to examine the shortened versions of the DASS-21 (DASS-13, DASS-12, DASS-9 (two versions), and DASS-8) for invariance across age and gender groups, discriminant validity, predictive validity, item coverage, and internal consistency, respectively. Compared with the DASS-21, all three-factor structures of the shortened versions expressed good fit, with the DASS-8 demonstrating the best fit and highest item loadings on the corresponding factors in both samples (χ^2^(16, 15) = 16.5, 67.0; p = 0.420, 0.001; CFI = 1.000, 0.998; TLI = 0.999, 0.997; RMSEA = 0.013, 0.059, SRMR = 0.0186, 0.0203). The DASS-8 expressed configural, metric, and scalar invariance across age and gender groups. Its internal consistency was comparable to other versions (α = 0.94). Strong positive correlations of the DASS-8 and its subscales with the DASS-21 and its subscales (r = 0.97 to 0.81) suggest adequate item coverage and good predictive validity of this version. The DASS-8 and its subscales distinguished the clinical sample from the general public at the same level of significance expressed by the DASS-21 and other shortened versions, supporting its discriminant validity. Neither the DASS-21 nor the shortened versions distinguished patients diagnosed with depression and anxiety from each other or from other psychiatric conditions. The DASS-8 represents a valid short version of the DASS-21, which may be useful in research and clinical practice for quick identification of individuals with potential psychopathologies. Diagnosing depression/anxiety disorders may be further confirmed in a next step by clinician-facilitated examinations. Brevity of the DASS-21 would save time and effort used for filling the questionnaire and support comprehensive assessments by allowing the inclusion of more measures on test batteries.

## 1. Introduction

Depressive and anxiety disorders are widespread in the general population, especially during the current COVID-19 pandemic [1,2]. Their increased occurrence during the COVID-19 crisis is due to the emotional reactions associated with the widespread nature of the disease, the grave adverse effects of the disease, as well as the lockdown adopted to protect against COVID-19, which entails restricted social interactions and loneliness implied by increased internet use as a defective coping method [2,3]. The pandemic is also associated with developing unhealthy dietary patterns and decreased levels of physical activity due to increased time spent at home [4,5,6]. These factors are associated with the development of a systematic inflammatory reaction that may affect brain regions involved in emotional regulation, resulting in the development of affective dysfunctions [7,8]. Moreover, SARS-CoV-2, the causative virus of COVID-19, causes neurodegeneration via direct invasion of brain cells and the cytokine storm, resulting in accelerated rates of the development of depression and anxiety in COVID-19 survivors [9]. In addition to their high occurrence in a wide-range of patient populations [10,11,12,13], depressive and anxiety disorders are also highly co-morbid with one another [14,15,16]. Their comorbidity is associated with common risk factors (e.g., childhood trauma and age of onset), and it coincides with the worst functional, somatic, and psychiatric outcomes [14]. 

Aggregate data denote a widespread prevalence of depression, anxiety, and low self-esteem among youth [17]. An exponential increase in this prevalence (up to 44%) has been witnessed during COVID-19, especially in the later years of the pandemic [18,19]. Females exhibit the highest prevalence of youth psychopathologies—almost twice the rate of males [18,19,20,21]. Youth psychopathologies are strongly associated with parental depression/anxiety, indicating a considerable genetic influence [20,22]. Depression, anxiety, and binge drinking in adolescents are associated with polymorphisms in the serotonin transporter (5-HTT) S-allele, the monoamine oxidase A (MAOA) low-activity alleles, and the dopamine D2 receptor (DDR2) Taq A1 allele [23].

The developmental psychology approach is an ecological–transactional model of development, which emphasizes the ontogenic and dynamically interacting aspects of psychological development [24]. Research has established the fetal phase of life as a sensitive period of development. Prenatal maternal substance use and maternal malnutrition as well as maternal stress can exert significant negative effects on the behavioral, brain, and psychopathological outcomes in the developing offspring [25,26,27]. Although the exact pathways involved in the development of psychopathology due to in utero exposure to stress is unclear, findings from animals studies suggest that the physiological alterations associated with maternal adversities trigger dysfunctional adaptations in the fetal hypothalamic-pituitary-adrenal axis: (a) transplacental passage of maternal cortisol to the fetus; (b) increased production of placental corticotropin-releasing hormone, which may enter the fetal circulation; and (c) maternal stress-induced effects on the sympathetic nervous system may result in vasoconstriction and decreased uteroplacental blood flow [27]. Interestingly, infants’ temperament, anxiety, and depression develop as a result of prenatal maternal symptoms of depression and anxiety, which occur in up 27% of pregnancies and contribute to poor maternofetal attachment, lack of initiation or early termination of breast feeding, and high levels of postpartum parenting stress. These effects were less common in women with a previously established diagnosis of depression or anxiety, signifying the stressful nature of anxiety and depression that evolve during pregnancy [13].

Maternal mood dysfunction may also influence the development of psychopathologies during early childhood due to factors of innate dysfunctional neuroregulatory mechanisms that develop during pregnancy, heritability, child exposure to maternal negative affect, cognitions, and dysfunctional behaviors, which may create a stressful context of a child’s life [28]. Gene-environment interactions play a pivotal role in early-life psychopathology. Childhood stress, trauma, and chronic exposure to domestic violence act as environmental factors that interact with genes conducive to prolonged activation of the stress response, resulting in increased vulnerability to depression and anxiety psychopathology [21,29]. Dysfunctional parenting behaviors represent one of the key environmental factors that are associated with higher psychopathology among adolescents [20,30]. For children of mothers with depression/anxiety psychopathologies, father involvement in child care, pattern and course of maternal mood dysregulation, and child characteristics may act as moderators to the risk of child psychopathology [28]. In line, a longitudinal investigation revealed that maternal anxiety and depression act as predictors of anxiety and depression in their adolescent children, especially female children with low adrenocortical reactivity. On the other hand, depression in youth with high adrenocortical reactivity is reported to significantly predict maternal depression [31]. Thus, the complex nature of the intergenerational transmission of depression and anxiety psychopathologies highlights these conditions as lifelong burdensome conditions for youth, their families, and the global community.

Poor quality of life, impaired academic/work performance, disturbed social life, extreme hopelessness, the development of dysfunctional eating patterns, negative religious coping, turning to alcohol and drug use as defective coping, and suicide are commonly reported drawbacks of these conditions [2,25,32]. Accordingly, prompt identification and management of depression and anxiety among youth are necessary to prevent a wide range of grave morbidities. Below the age of 20 years, depression and anxiety as clinical diagnoses of emotional disorders are moderately co-morbid [33,34]. Among youths diagnosed with at least one emotional disorder, depression and anxiety as symptoms are more discriminable (i.e., by multiple factors) than among mental disease-free youth who express symptoms by a single factor [33].

The tripartite model has been proposed to discretely identify depression from anxiety and stress. The latter co-occurs in both conditions [35]. However, research indicates that depression and anxiety are more interrelated than previously thought. A meta-analysis involving 226 task-related functional imaging studies reports shared abnormalities (mainly hypoactivation) in task-related brain activation in regions primarily associated with inhibitory control and cognitive processing [16]. In fact, depression and anxiety are common prodromal symptoms in cognitive disorders associated with genetic tendencies such as Alzheimer’s disease [11]. Aggregate data pinpoint genetic and causal associations between major depression and anxiety disorders, suggesting that certain types of anxiety (e.g., post-traumatic stress) may represent subtypes of depressive disorders [36]. In fact, childhood anxiety seems to be influenced by a single genetic factor that does not contribute to genetic variance in depression symptoms. However, in adolescents and young adults, genetic influences are significantly shared between depression and all anxiety symptoms, along with a small significant genetic fear factor [34].

The Depression Anxiety Stress Scale (DASS) 42 and its short version (DASS-21) have been designed to match the tripartite model by differentiating the distinct features of depression, anxiety, and stress from each other [15]. However, subsequent tests show excessive variations in the structure of the DASS-21. While the three-factor structure of the DASS-21 is generally supported, a quadripartite structure involving three specific factors (depression, anxiety, and stress) and a general factor of emotional negativity or overall distress has been reported in several studies [37,38,39]. A tripartite model comprising anhedonia, physiological hyperarousal, and general negative affect had a better fit than other structures in the general public and a psychiatric sample in Turkey [40]. A two-factor structure involving depression and anxiety/stress factors expressed the best fit among Brazilian adolescents [41]. Many studies showed good fit of a one-factor structure of the DASS-21 [15,42,43,44,45,46].

Numerous studies investigated invariance of the DASS-21. Around half the items of the scale expressed differential item functioning (DIF) across gender and age groups among Egyptian drug users while only item 6 expressed DIF across gender groups among Iranian medical students [47]. In a sample of athletes, the bifactor structure of the DASS-21 was invariant across groups of gender, athletic expertise, sport type, and injury status [37]. On the contrary, in a multinational study, the bifactor structure of the DASS-21 was variant across different countries. Instead, the authors suggested the use of the scale as a unidimensional measure instead of being a measure of depression, anxiety, and stress [42]. Although the DASS-21 held invariance across respondents from the USA and the UK, it showed threshold invariance indicated by higher depression scores among Russian and Polish respondents. Compared with English-speaking respondents, Russian respondents exhibited the highest levels of anxiety symptoms while Polish respondents exhibited the highest stress levels [48]. Scalar variance between Pakistani and German university students was noted, with Pakistani students experiencing more symptoms of depression and anxiety [49].

Because the DASS-21 is not a clinical diagnostic measure, it is frequently used in research and practice in clinical and non-clinical samples in order to identify individuals with high distress who may be prone to develop psychopathologies. Its simplicity, brevity, and ability to capture symptoms of stress along with those of depression and anxiety make it more favorable than other specific measures of depression or anxiety [50,51]. The scale demonstrates good internal consistency and exhibits sensitivity to change following treatment (e.g., of depression). However, in addition to problems of non-invariance and structural variations, it persistently demonstrated a ceiling effect in three samples of depressed patients [52]. A ceiling effect is a key measurement error entailing scale attenuation effect that results from clustering of respondent scores around the highest possible score limit, which precludes variance estimation resulting in measurement inaccuracy [53]. Extending the response scale to include an additional option did not abolish the ceiling effect, suggesting a need for extensive revision of the scale [52].

Few studies revised the DASS-21 structure. Employing item response theory and confirmatory factor analysis (CFA), Osman and colleagues suggested that 13 or nine items may best reflect the three distinct structures of the DASS-21 in non-clinical samples [51]. Seventeen items were reported to better capture the distress component covered by the DASS-21 among Egyptian drug users [15]. A subsequent investigation reported usability of a 12-item DASS based on Osman’s model in Korean psychiatric patients and the general public [32]. However, none of these structures has been tested in other studies signifying the DASS-21 as the official short form of the DASS-42.

We have previously evaluated the structure of the DASS-21 among drug users, and the scale turned out to best describe overall distress instead of differentiating the constructs of depression, anxiety, and stress [15,44]. However, drug users represent a population that express problems with emotional regulation, which may affect the manner through which they can express different aspects of emotional negativity [54,55]. In the meantime, the DASS-21 has been widely used as a measure of mental health symptomology both in healthy and vulnerable groups during the COVID-19 crisis [56]. Therefore, it may be necessary to evaluate the structure of the Arabic DASS-21 in other clinical samples as well as in the general public who express varying levels of emotional negativity [32,57]. In addition, standard Arabic is not easy to understand in Egypt, especially among people with low levels of education. This is because the Arabic language in Egypt has been drastically altered over the long periods of occupation that Egypt has witnessed in its recent history (e.g., by Turkey, France, UK, and Israel). Therefore, the validated Arabic version of the DASS-21, which is available in the local Egyptian accent may not be easy to understand in other Arab countries where the local accent is closer to standard Arabic—the most collective and comprehendible form [58,59]. The present research aims to fill this gap by examining the internal consistency, factor structure, invariance, and discriminant validity of a standard Arabic version of the DASS-21 among psychiatric patients and the general public. It also tests the psychometric properties of different shortened versions of the DASS-21.

## 2. Materials and Methods

### 2.1. Study Design, Participants, and Procedure

This cross-sectional study is a secondary analysis of data based on two convenient samples. The first sample [60], herein referred to as the quarantine sample, comprised 214 Saudi citizens or residents of Saudi Arabia who were quarantined for 14 days in seven quarantine facilities in the cities of Riyadh and Qassim. People undergoing quarantine were travelers returning to Saudi Arabia during the lockdown period as well as suspected or confirmed COVID-19 cases with mild disease. Participants were included in the study if they were 18 years or older, could speak Arabic, and agreed to participate in the study. Data were collected via an online survey during the period between 29 April and 19 May 2020, since direct contact was strongly prohibited by the Saudi authorities.

The second sample was obtained via an anonymous online survey distributed via Twitter and WhatsApp groups in Saudi Arabia during April 2020—the beginning of the formal confinement period in the country. Respondents testifying that their age was above 18 years who signed a digitized informed consent form were directed to the online questionnaire. Among 1160 respondents, 168 reported having a preexisting mental disorder which was diagnosed by a psychiatrist. Accordingly, respondents were classified based on the criteria of having or not having a psychiatric diagnosis into two samples: a psychiatric patient sample (sample 1) and a community sample (sample 2).

### 2.2. Study Instruments

Our respondents were presented a structured online questionnaire that comprised several sections (described in detail elsewhere). In brief, the first section involved assessment of sociodemographic, clinical, and COVID-19-related data, e.g., age, education, health status, having a physical disease or a psychiatric disorder, and views on COVID-19. The second section comprised the standard Arabic version of the Depression Anxiety Stress Scale-21 (DASS-21), which was obtained from the official website of the DASS [http://www2.psy.unsw.edu.au/DASS/Arabic/Arabic%20DASS-21.pdf (access on 25 January 2020)], with less information available on its psychometric properties. The DASS-21 contains 21 items in three subscales, which assess symptoms of depression (items 3, 5, 10, 13, 16, 17, 21), anxiety (items 2, 4, 7, 9, 15, 19, 20), and stress (items 1, 6, 8, 11, 12, 14, 18) [15]. The degree to which respondents endorsed the symptoms over the course of the last week is rated on a scale that ranges from 0 (did not apply to me at all) to 3 (applied to me very much or most of the time). Higher scores reflect higher levels of symptom endorsement [44]. Reliability of the DASS-21, as evaluated in the quarantine sample, sample 1, and sample 2, is excellent (α = 0.95, 0.96, and 0.94, respectively).

### 2.3. Ethical Considerations

The Institutional Review Board of Al Qassim University approved the study protocol (No. 19-08-01). Potential respondents were introduced to a digital consent form emphasizing that participation was voluntary, and that data were anonymously collected, confidential, and would only be used for scientific purposes.

### 2.4. Statistical Analysis

In the quarantine sample, exploratory factor analysis (EFA) involving maximum-likelihood extraction and varimax rotation with the Kaiser-Meyer-Olkin (KMO) measure of sampling adequacy and Bartlett’s test of sphericity was used to let items of the DASS-21 freely load on the corresponding factors without enforcing any constraints. In a next step involving sample 1 and sample 2, CFA, with maximum likelihood and bootstrapping involving 2000 random samples, was used to check data-fit to various models. In this study, we tested 13 competing models: Model 1, a one-factor structure; Model 2, a two-factor structure comprising depression and anxiety/stress factors; Model 3, Lovibond’s original three-factor structure; Model 4, a bifactor structure (a general factor and three specific factors); Model 5, the previously reported 17-item one-factor structure; Model 6 and Model 7, a one-factor and a three-factor structure based on the Korean 12-item DASS [32]; Model 8 and Model 9 as well as Model 10 and Model 11, a one-factor and a three-factor structure based on a 13-item and a 9-item DASS suggested by Osman and colleagues [51]; Model 12 and Model 13, a three-factor structure of a modified 9-item DASS and an 8-item DASS based on eliminating items with lower loadings and item-total correlations.

Global model fit was flagged by a non-significant chi square (χ^2^) index [61]. However, χ^2^ is sample size-dependent [62]. Therefore, good and acceptable fit were decided based on absolute fit indices: Comparative Fit Index (CFI) and Tucker–Lewis Index (TLI) equal to or above 0.95 and 0.90, respectively, along with root mean square error of approximation (RMSEA) and standardized root-mean-square residual (SRMR) less than 0.06 and 0.08, respectively [15,63]. For all models, modification indices were consulted, and improvements in model fit following correlating suggested error residuals were recorded.

To examine measurement invariance of the shortened versions of the DASS-21 across groups of gender and age (30 years old and below; above 30 years), multigroup CFA was used. The analysis comprised four models. The first model was unconstrained, and it tested the overall fit (same number of factors) of the shortened versions across groups, known as configural invariance. The second model constrained factor loadings to equality between groups and evaluated metric invariance as a function of the difference between the unconstrained and constrained model. The third model assessed scalar invariance (scale mean differences) by constraining the intercepts of the items to be equal between groups. The fourth model tested strict invariance by constraining the residuals to be equal between groups [3,63]. Although χ^2^ may reflect changes in model fit across groups, it is sample-size dependent to a great extent—unlike absolute model fit indicators such as CFI and RMSEA. Therefore, we depicted invariance across subgroups by significant changes in CFI and RMSEA—for invariance, ΔCFI and ΔRMSEA should not exceed 0.02 and 0.015, respectively [62].

Normality of the DASS-21 and its shortened versions was tested by Shapiro–Wilks’ W test. Internal consistency of the DASS-21, its subscales, as well as the shortened versions and their subscales was assessed by coefficient alpha, alpha-if-item deleted, and item-total correlations. Correlations between the shortened versions and their subscales with the DASS-21 and its subscales was used to signify item coverage and predictive validity of the best fitting shortened version of the DASS-21. Because of the non-normal distribution of the DASS-21 and all its shortened versions, the Mann Whitney U test was used to examine discriminant validity of the best fitting shortened version of the DASS-21 by comparing the mean of the scale, as well as of the depression, anxiety, and stress subscales in the samples. To identify if the depression and anxiety subscales on the DASS-21 and its shortened versions can differentiate people with depression and anxiety from those with other disorders, the Mann Whitney U test was used to compare the mean of depression and anxiety among patients with and without depression and with and without anxiety, respectively. All analyses were conducted in SPSS and Amos, and significance was considered at a probability level less than 0.05, two-tailed.

## 3. Results

### 3.1. Participants’ Characteristics

For the quarantine sample, males were a majority (59.8%), 49.5% of the participants were married while 49.1% were single, most participants were in the age groups 18–30 years (56.1%) and 31–40 years (30.4%), most participants had a bachelor degree (41.1%) or a masters degree (23.8%), and being a student was the most common employment form (45.3%), while 39.3% had other forms of employment. The reported family income ranged from less than 5000 Saudi Rial (SAR, 16.4%) to more than 25,000 SAR (15.9%). However; the family income of 47.2% of the participants ranged between 5000 and 15,000 SAR. SAR equals 0.27 US dollar.

The clinical sample (sample 1) comprised patients with psychiatric disorders (N = 168). Key reported diagnoses were depression (40.5%), generalized anxiety disorder (41.7%), sleep disorders (23.8%), and obsessive compulsive disorders (OCD, 15.5%). Comorbidity was common, especially of sleep disorders and OCD among patients with anxiety and depression. In addition, 36.3% of the participants reported other disorders such as eating disorders, post-traumatic stress disorder, personality disorders, bipolar disorder, and psychotic disorders. The community sample (sample 2) comprised 992 respondents with no reported psychiatric diseases. For sample 1 and sample 2, in order, most participants were females (70.8% and 62.7%), were aged 31 years and above (48.2% and 54.8%) while 51.8% and 45.2% were in the age category 18-30 years. As for the educational level, 62.5% and 61.0% of the participants had a university degree while 19.0% and 15.2% had high school. Those employed, unemployed, retired, and students represented 29.2%, 26.8%, 6.0%, and 38.1% of the participants in sample 1 and 41.1%, 22.1%, 10.4%, and 27.4% of the participants in sample 2. The reported family income ranged from less than 5000 SAR (11.0% and 7.2%) to more than 25,000 SAR (16.7% and 17.1%). However; the family income of 44.6% and 44.0% of the participants ranged between 5000 and 15,000 SAR. See Appendix A for further details of the sociodemographic characteristics of all the samples.

### 3.2. Results of Exploratory Factor Analysis

EFA revealed that the DASS-21 in the quarantine sample covers four factors with eigen values >1, which explained 48.3%, 7.0%, 5.8%, and 4.9% of the variance. The sample size and participant-to-item ratio were appropriate for EFA: KMO values = 0.924, Bartlett’s test was significant (χ^2^(210) = 2887.78, *p* < 0.001). As shown in Table 1, several items loaded on two factors with loadings greater than 4. Item communalities, scree plots, and reproduced correlations are presented in Appendix A.

### 3.3. Results of Confirmatory Factor Analysis

Examination of different structures of the DASS-21 (unidimensional, two-factor, and three-factor) revealed poor fit in crude models. Acceptable fit was achieved by correlating a few item residuals in sample 1 and several item residuals in sample 2 (Table 2). The bifactor structure of the DASS-21 expressed acceptable fit, with all items loading significantly on the common factor, but none of the item loadings on the anxiety factor were significant. SRMR was not calculated in the bootstrapped model, signifying a problem with the fit of this model, and when the iteration limit was increased, the model failed to converge. As for the shorter versions of the DASS-21, the 17-item structure previously tested among Egyptian drug users [15] expressed poor fit in both samples. Acceptable fit of this structure was produced by correlating item 19 with item 4 and item 20 with item 15 in sample 1 and numerous items [(1 with 3 and 12), (17 with 8 and 10), and (4 with 19)] in sample 2. The crude one-factor structures of Osman’s DASS-13, the Korean DASS-12, and Osman’s DASS-9 expressed acceptable fit mostly in sample 1, and correlating few items improved the fit in sample 2. The three-factor structure of the DASS-13, DASS-12, and Osman’s DASS-9 had excellent fit in both samples—correlating few items in sample 2 was necessary to improve the fit in most models (Figure 1). Noticeably, our crude DASS-9/DASS-8 models expressed superior fit in sample 1 while correlating few error terms considerably improved model fit in sample 2. Nonetheless, the fit of the DASS-8 with correlated residuals expressed a perfect fit in both samples (Table 2). As shown in Figure 1, the item loadings on the corresponding factors in the DASS-8 were greater than in all other shortened versions, implying that the DASS-8 describes the best fit of the data in both samples.

### 3.4. Results of Invariance Analysis

Multigroup analysis revealed invariance of all shortened versions of the DASS across gender groups (Appendix A). The same goes for age groups; however, the DASS-9 based on Osman’s analysis expressed significant variance at the scalar level across age groups in sample 2 (χ^2^(56) = 208.3, *p* < 0.001, Δχ^2^ = 91.1, p(Δχ^2^) = 0.001, ΔCFI = 0.032, ΔTLI = 0.38, ΔRMSEA = −0.015). All shortened versions of the DASS-21 expressed variance at the strict level across age groups (Appendix A).

### 3.5. Normality of the DASS-21 and Its Shortened Versions

Values of the Shapiro–Wilks’ W suggest that the normality of our DASS-9, the DASS-8, and their subscales is comparable with that of the DASS-21 and its subscales in both samples, Table 3. It was also comparable with other shortened versions of the DASS-21 (Appendix A).

### 3.6. Internal Consistency, Item Coverage, and Predictive Validity of Shortened Versions of the DASS-21

The DASS-21 and all other shortened versions expressed good internal consistency in the samples. The reliability of our DASS-9/DASS-8 was higher than the previously tested Korean 12-item DASS and Osman’s 13- and 9-item DASS. As shown in Table 4, the DASS-9/DASS-8 expressed the highest item-total correlations in both samples compared with all other versions. In sample 1, the correlations between the DASS-9/DASS-8 and the DASS-21 were high comparable with the DASS-12 (α = 0.97). In sample 2, the correlations between the DASS-9/DASS-8 and the DASS-21 were a bit lower than that of the DASS-12; however, they were still high (α = 0.95 and 0.93).

As shown in Table 5, internal consistency of subscales of the DASS-8 were comparable with all other shortened versions. The anxiety subscale on the modified DASS-9/DASS-8 had even higher reliability than the anxiety original subscale. Its correlation with the DASS-21 and the original anxiety subscale was higher than the correlations expressed by all the anxiety subscales of other shortened versions. Meanwhile, the correlation of the depression subscale with the DASS-21 and its depression subscale was comparable with those of the Korean DASS-12. Although it comprises half the number of items on the stress subscale of the Korean DASS-12, the stress subscale on the DASS-9/DASS-8 expressed strong significant correlations with the DASS-21 and its stress subscale. Altogether, the high cross-scale correlations between the DASS-9/DASS-8 and their subscales with the DASS-21 and its subscales suggest an acceptable coverage and an almost similar predictive validity of the DASS-9/DASS-8 to the DASS-21.

### 3.7. Discriminant Validity of the DASS-21 and Its Shortened Versions

The Mann Whitney U test revealed that the DASS-21 and all shortened versions as well as their subscales (Table 4 and Table 6) could differentiate the clinical sample from the general public (all *p* values < 0.001). However, the depression and anxiety subscales on the DASS-21 and on all the shortened versions could not differentiate patients diagnosed with depression or anxiety from patients having other psychiatric diagnoses (all *p* values > 0.05, Appendix A).

## 4. Discussion

This study examined the psychometric properties of a standard Arabic version of the DASS-21 as well as five shortened versions of the DASS-21 in a quarantined sample because of COVID-19, a clinical sample, and in the general public through various robust testing techniques. The unidimensional, tripartite, and quadripartite structures of the DASS-21 involving correlated errors expressed acceptable fit in both samples denoting usability of the overall score of the scale as well as its subscales. Among different shortened versions of the DASS-21, the DASS-8 expressed the best fit and the highest item loadings on the corresponding factors, along with invariance across age and gender groups.

EPA revealed a four-factor structure of the DASS-21 in the quarantine sample. Two factors and three factors with eigen values >1 were also produced in sample 1 and sample 2, respectively (Appendix A). In the all the samples, a large number of items had significant cross-loadings on several items (Table 1 and Appendix A). This finding indicates that the supposed causal contribution of indicators (i.e., factor loadings) to the underlying factors are not correctly specified, which is consistent with former studies calling for revising the item structure of the DASS-21 [51,52]. Developing a valid shortened form of a scale should be guided by both statistical and content approaches [63]. For reducing the DASS-21 in this study, we employed two samples to build three models based on previous studies that revised the item structure of the DASS-21 [32,51]. In addition, we have stepwise removed items with relatively low loadings and low item-total correlations within the DASS-21 and its subscales. However, to decide on items to be retained among many items with adequate loadings/item-total correlations, we examined the descriptive statistics of all items of the DASS-21, highlighting those with the lowest means and highest SDs in both samples. We then referred to reports in the literature on the frequency of reported items.

Regarding the depression subscale, examinations of corrected item total correlations of Osman’s DASS-9 in both samples revealed lower values for items reflecting on worthlessness and hopelessness “item 17, I felt I wasn’t worth much as a person” and “item 21, I felt that life was meaningless” than those for items reflecting on depressed mood and lack of motivation/psychological fatigue “item 13, I felt down-hearted and blue” and “item 16, I was unable to become enthusiastic about anything”, respectively. It may be intuitive that the general public are less likely to experience worthlessness and hopelessness symptoms, which may be more evident in individuals with manifest and severe depression. Feeling worthless and hopeless are key symptoms that can mostly differentiate depressed from nondepressed patients [64]. These symptoms also strongly correlate with suicide ideation [65]. However, investigations reporting on the frequency of depressive symptoms in different conditions (e.g., traumatic brain injury, the general public, nursing students) report higher prevalence of symptoms of fatigue, anhedonia, insomnia, and severe feelings of sadness or depressed mood [64,65,66]. In an investigation involving 117 patients with partially or fully remitted major depressive disorder, fatigue was highly associated with feeling “blue”. Both symptoms, along with lack of interest were associated with symptoms of inability to focus, alertness, and difficulty concentrating. Cognitive deficit was not associated with symptoms of self-blame, feeling worthless, feeling hopeless, suicidal thoughts, sleep difficulty, and lack of appetite [67]. Therefore, we have restructured the depression subscale on our DASS-9 by replacing item 17 and item 21 with item 13 and item 16. This change has increased the reliability of the depression subscale as well as overall reliability of the DASS-9 in both samples. Noting that the loading of item 3 was comparatively lower than other items, we have removed it, with no subsequent reduction in the reliability of our resulting DASS-8. The discriminant validity of this three-item depression subscale was not altered, as noted below.

As for the anxiety subscale on Osman’s DASS-9, item 2 and item 4 had the lowest item-total correlations (0.500 and 0.522 among patients; 310 and 333 among the general public). On the anxiety subscale of the DASS-21, those two items had lower item total correlations than other items. The frequency of their occurrence along with physical symptoms of anxiety (e.g., trembling hands and dry mouth) was low. Dry mouth is reported in only 20% of patients with GAD. GAD patients demonstrate high peripheral catecholamine levels when in a resting state as well as a blunted sympathetic response to acute stress secondary to psychopathological responsiveness of the sympathetic adrenal medulla system [68]. The same physiological alteration has been reported in first-episode, drug-naïve patients with panic disorder [69]. Circulating catecholamines are not associated with the acute increase in heart rate during panic attacks [70]. These reports might justify why participants in both samples reported item 19 “less aware of the action of my heart” at a frequency lower than that of item 9 “worried about situations in which I might panic”, 15 “felt I was close to panic” and 20 “felt I was scared without a good reason”, which are all relevant to the hypervigilant experience of panicking. Accordingly, we have replaced item 2, 4, and 19 with item 9, 15, and 20. As shown in Figure 1, these items had considerably higher loadings on the anxiety factor in models representing our DASS-9/DASS-8. In addition, the reliability of this three-item anxiety subscale in both samples was higher than all other anxiety subscales, including that of the original DASS-21—its reliability in the general public sample was similar to that of the anxiety subscale of the DASS-21. Its correlation with the DASS-21 and the original anxiety subscale was the highest relative to all the shortened anxiety subscales (Table 5), granting it the highest predictive validity.

Because the loading of item 1 on the stress subscale of Osman’s DASS-9 was lower than that of item 8 “I was using a lot of my nervous energy” on Osman’s DASS-13, we retained item 8 on the DASS-9/DASS-8 instead of item 1, ending with a subscale that comprises items 8 and 12 only. The response to life stresses frequently involves an intrusive state represented by symptoms of unbidden ideas and feeling [71,72]. Encountering these symptoms can deplete the psychic energy resulting in a state of mental exhaustion [72]. Difficulty relaxing “item 12” is commonly reported in people undergoing stress e.g., musicians with hearing difficulties [73]. The reliability of the two-item stress subscale of the DASS-9/DASS-8 was comparable with that of the four-item stress subscale of the Korean DASS-12 (0.835 versus 0.852). Its correlation with the DASS-21 and its stress subscale was also comparably strong (Table 4), denoting adequate predictive validity of this shortened subscale.

Although the DASS-8 comprises only two thirds the number of items on the previously tested Korean DASS-12, it expressed higher internal consistency, higher item total correlations, and similarly strong correlations with the original DASS-21 in both samples (Table 3, Table 4 and Table 5). Its correlation with the DASS-21 in sample 2 was strong but a bit lower than the Korean version. Reducing items on a symptom scale that comprises multiple replicate items may decrease its reliability, sensitivity or specificity if it involves items with optimal or close to optimal sensitivity and specificity. On the other hand, dropping heterogeneous items would increase the reliability of the scale [74]. Thus, the results suggest that items on the DASS-8 and its subscales, as discussed above, possess optimal sensitivity and specificity, implying adequate item coverage and relatively good predictive validity of the DASS-8 than other shortened versions of the DASS-21.

The Mann Whitney W test revealed that the DASS-8 could differentiate psychiatric patients from the general public at the same level of significance of all the shortened versions of the DASS-21 (Table 3). On the other hand, the depression subscale on the DASS-21 and all the shortened versions of the DASS-21 could not differentiate patients with depression from those with other psychiatric disorders. The same goes for the anxiety subscale. In previous studies, the DASS-21 [12,40,75] and the DASS-12 [32] could only distinguish healthy participants from those with psychopathologies that include both symptoms of depression and anxiety. However, the DASS-21 could not differentiate people with depressive disorder from those with anxiety disorder [12,40]. Because the DASS-21 is not a clinical diagnostic tool [50], it may be beneficial for screening large groups for the possibility of encompassing psychopathologies, which may be confirmed by further investigations. In this respect, the DASS-8 may be an ideal short form of the DASS-21 for initial identification of distressed individuals. Its configural, metric, and scalar invariance across age and gender groups in the clinical sample and in the general public support its usability as a valid measure of symptoms of distress in various groups. Further investigations of the DASS-8 in different populations are needed.

This study enjoys the merit of being the first to extensively reduce the DASS-21 to a valid and reliable 8-item version that expresses an adequate ability to measure symptoms of distress objectively across ages and sexes as well as to differentiate individuals with psychopathology from healthy individuals. It also tested a standard Arabic version of the DASS-21, which can be generally used in all Arab countries. Several limitations to the generalizability of our results should be also acknowledged. Collecting data through a self-administered questionnaire and an online survey method entails risks for social desirability bias and selection bias. The cross-sectional design precluded test-retest reliability analysis. The clinical sample was established based on participants’ subjective reports of receiving a psychiatric disorder diagnosis by a psychiatrist instead of being screened for psychopathology according to a known disease classification system (e.g., DSM-IV-R). In the meantime, lack of screening of respondents in the community sample for mental disorders entails that some of those respondents may not be free of mental illness. This may cast doubt on the soundness of the comparisons of the psychometrics of measures, particularly discriminant validity, of the DASS-21 and its shortened versions between the two samples. In addition, the numerical imbalance between the two samples is another considerable limitation. Moreover, the samples came from a single Arab country, while the DASS-21 is reported to express invariance at the configural [42] and scalar [48,49] levels across countries. Therefore, examining the psychometric properties of the DASS-8 in other countries/languages is necessary for effective usability of the scale in clinical practice and research.

## 5. Conclusions

The scores of the DASS-21 and its subscales may be used to reflect on symptoms of distress. However, compared with four other shortened versions of the DASS-21, an 8-item version (DASS-8) demonstrated perfect fit, measurement invariance across age and gender groups, adequate item coverage, good predictive validity, and excellent internal consistency. The DASS-8 differentiated patients with psychiatric disorders from the general public at the same level of significance exhibited by the DASS-21 and shortened versions that comprised more items. Thus, the DASS-8 is a brief tool that can be used in clinical practice and research to facilitate the detection of psychopathologies and monitor response to treatments at the symptom level. Further evaluations of the DASS-8 in diverse populations are necessary for optimal usage of the scale.

## Figures and Tables

**Figure 1 ijerph-18-10142-f001:**
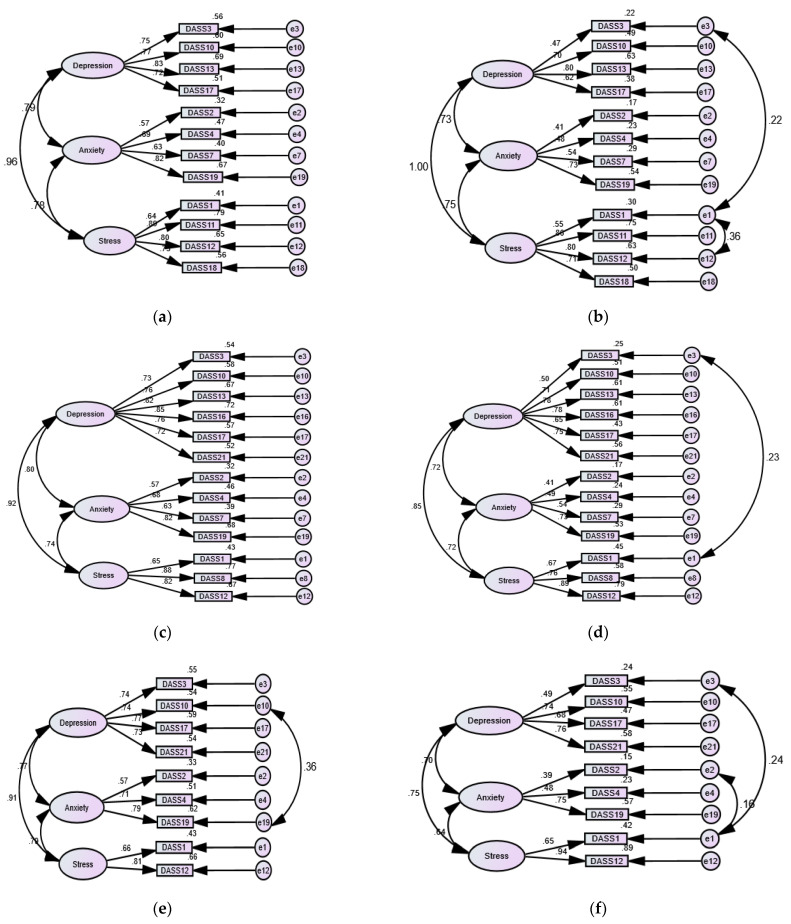
Three-factor structures of shortened versions of the Depression Anxiety stress scale 21 (DASS-21) in patients with psychiatric disorders and in the general public, in order: the DASS-12 (**a**,**b**), the DASS-13 (**c**,**d**), Osman’s DASS-9 (**e**,**f**), a modified DASS-9 (**g**,**h**), and DASS-8 (**i**,**j**).

**Table 1 ijerph-18-10142-t001:** Item loadings on corresponding factors as revealed by exploratory factor analysis of the Depression Anxiety Stress Scale 21 (DASS-21) in the quarantine sample.

Items		Extracted Factors			
		Factor 1	Factor 2	Factor 3	Factor 4
1	I found it hard to wind down	0.242	0.178	0.206	**0.616**
2	I was aware of dryness of my mouth	**0.573**	0.182	0.149	0.136
3	I couldn’t seem to experience any positive feeling at all	0.265	0.278	**0.365**	0.284
4	I experienced breathing difficulty (e.g., excessively rapid breathing, breathlessness in the absence of physical exertion)	**0.451**	0.162	0.285	0.078
5	I found it difficult to work up the initiative to do things	0.246	0.206	**0.474**	0.022
6	I tended to over-react to situations	0.251	**0.587**	0.194	0.141
7	I experienced trembling (e.g., in the hands)	**0.584**	0.320	0.121	0.225
8	I felt that I was using a lot of nervous energy	0.257	**0.659**	0.270	**0.439**
9	I was worried about situations in which I might panic and make a fool of myself	0.345	**0.695**	0.160	0.179
10	I felt that I had nothing to look forward to	0.124	0.244	**0.604**	0.205
11	I found myself getting agitated	0.348	**0.552**	0.390	**0.427**
12	I found it difficult to relax	0.250	0.364	0.258	**0.800**
13	I felt down-hearted and blue	0.293	**0.531**	**0.531**	0.194
14	I was intolerant of anything that kept me from getting on with what I was doing	0.100	**0.496**	**0.465**	0.315
15	I felt I was close to panic	**0.745**	0.331	0.181	0.292
16	I was unable to become enthusiastic about anything	0.342	0.166	**0.701**	0.267
17	I felt I wasn’t worth much as a person	0.364	0.156	**0.555**	0.257
18	I felt that I was rather touchy	0.298	**0.562**	0.389	0.151
19	I was aware of the action of my heart in the absence of physical exertion (e.g., sense of heart rate increase, heart missing a beat)	**0.624**	0.092	0.312	0.092
20	I felt scared without any good reason	**0.699**	0.296	0.234	0.223
21	I felt that life was meaningless	**0.548**	0.257	**0.461**	0.160

Values in boldface represent significant loadings.

**Table 2 ijerph-18-10142-t002:** Goodness-of-fit indices for different models of the Depression Anxiety Stress Scale 21 (DASS-21) and its shortened versions evaluated by confirmatory factor analysis.

Models	Samples	χ^2^	*P*	*Df*	CFI	TLI	RMSEA	RMSEA 90% CI	SRMR
Model 1 1F DASS-21	Sample 1 (C)	433.259	0.000	189	0.902	0.891	0.088	0.077 to 0.099	0.0504
	Sample 2 (C)	1399.543	0.000	189	0.889	0.877	0.080	0.076 to 0.084	0.0464
	Sample 1 (E)	351.518	0.000	186	0.934	0.925	0.073	0.061 to 0.085	0.0460
	Sample 2 (E)	964.340	0.000	183	0.929	0.918	0.066	0.062 to 0.070	0.0412
Model 2 2F DASS-21	Sample 1 (C)	410.409	0.000	188	0.911	0.901	0.084	0.073 to 0.095	0.0498
	Sample 2 (C)	1305.573	0.000	188	0.898	0.886	0.077	0.074 to 0.081	0.0447
	Sample 1 (E)	328.718	0.000	185	0.942	0.935	0.068	0.056 to 0.080	0.0452
	Sample 2 (E)	819.684	0.000	179	0.941	0.931	0.060	0.056 to 0.064	0.0373
Model 3 3F DASS-21	Sample 1 (C)	396.040	0.000	186	0.916	0.905	0.082	0.071 to 0.093	0.0489
	Sample 2 (C)	1205.022	0.000	186	0.907	0.895	0.074	0.070 to 0.078	0.0427
	Sample 1 (E)	360.727	0.000	184	0.929	0.919	0.076	0.064 to 0.087	0.0460
	Sample 2 (E)	818.366	0.000	179	0.942	0.931	0.060	0.056 to 0.064	0.0366
Model 4 Bifactor DASS-21	Sample 1 (C)	838.898	0.000	184	0.938	0.929	0.071	0.059 to 0.083	--
	Sample 2 (C)	1207.798	0.000	184	0.906	0.893	0.075	0.071 to 0.079	--
Model 5 1F DASS-17 item	Sample 1 (C)	311.184	0.000	119	0.905	0.898	0.098	0.085 to 0.112	0.0510
	Sample 2 (C)	1957.672	0.000	119	0.892	0.877	0.089	0.084 to 0.094	0.0477
	Sample 1 (E)	249.932	0.000	117	0.934	0.924	0.082	0.068 to 0.097	0.0458
	Sample 2 (E)	329.896	0.000	60	0.948	0.932	0.067	0.060 to 0.075	
Model 6 1F Korean DASS-12	Sample 1 (C)	161.031	0.000	65	0.924	0.908	0.094	0.076 to 0.112	0.0561
	Sample 2 (C)	598.906	0.000	65	0.903	0.884	0.091	0.084 to 0.098	0.0529
	Sample 1 (E)	139.024	0.000	64	0.940	0.927	0.084	0.065 to 0.103	0.0522
	Sample 2 (E)	399.349	0.000	62	0.939	0.923	0.074	0.067 to 0.081	0.0465
Model 7 3F Korean DASS-12	Sample 1 (C)	81.966	0.004	51	**0.971**	**0.962**	**0.060**	0.034 to 0.084	0.0388
	Sample 2 (C)	375.549	0.000	51	0.930	0.910	0.080	0.073 to 0.088	0.0423
	Sample 2 (E)	214.140	0.000	49	**0.965**	**0.952**	**0.058**	0.050 to 0.066	0.0348
Model 8 1F Osman’s DASS-13	Sample 1 (C)	154.117	0.000	65	0.927	0.912	0.091	0.072 to 0.109	0.0560
	Sample 2 (C)	585.667	0.000	65	0.899	0.879	0.090	0.083 to 0.097	0.0535
	Sample 1 (E)	133.026	0.000	64	0.943	0.931	0.080	0.061 to 0.100	0.0525
	Sample 2 (E)	329.896	0.000	60	0.948	0.932	0.067	0.060 to 0.075	0.0442
Model 9 3F Osman’s DASS-13	Sample 1 (C)	95.869	0.004	62	**0.972**	**0.965**	**0.057**	0.033 to 0.079	0.0413
	Sample 2 (C)	297.251	0.000	62	0.954	0.943	0.062	0.055 to 0.069	0.0380
	Sample 2 (E)	251.989	0.000	61	**0.963**	**0.953**	**0.056**	0.049 to 0.064	0.0347
Model 10 Osman’s DASS-9	Sample 1 (C)	80.404	0.000	27	0.917	0.889	0.106	0.082 to 0.137	0.0585
	Sample 2 (C)	366.421	0.000	27	0.875	0.833	0.113	0.103 to 0.123	0.0586
	Sample 1 (E)	64.291	0.000	26	0.940	0.917	0.094	0.065 to 0.123	0.0535
	Sample 2 (E)	162.073	0.000	25	0.938	0.911	0.082	0.072 to 0.093	0.0505
Model 11 Osman’s DASS-9	Sample 1 (C)	53.590	0.000	27	0.954	0.931	0.086	0.055 to 0.117	0.0468
	Sample 2 (C)	366.421	0.000	27	0.875	0.833	0.113	0.103 to 0.123	0.0586
	Sample 1 (E)	41.339	0.011	23	**0.971**	**0.955**	**0.069**	0.033 to 0.102	0.0419
	Sample 2 (E)	72.478	0.000	22	**0.981**	**0.969**	**0.048**	0.036 to 0.061	0.0283
Model 12 A modified 3F DASS-9	Sample 1 (C)	38.653	0.030	24	0.987	0.980	0.060	0.019 t0 0.094	0.0302
	Sample 2 (C)	225.860	0.000	24	0.955	0.933	0.092	0.081 to 0.103	0.0323
	Sample 1 (E)	**24.835**	**0.359**	23	**0.998**	**0.997**	**0.022**	0.000 to 0.069	0.0219
	Sample 2 (E)	134.101	0.000	23	**0.976**	**0.962**	**0.070**	0.059 to 0.081	0.0265
Model 13 3F DASS-8	Sample 1 (C)	30.003	0.026	17	0.987	0.979	0.068	0.023 to 0.107	0.0275
	Sample 2 (C)	217.990	0.000	17	0.953	0.923	0.109	0.097 to 0.122	0.0351
	Sample 1 (E)	**16.483**	**0.420**	16	**1.000**	**0.999**	**0.013**	0.000 to 0.073	**0.0186**
	Sample 2 (E)	**67.047**	**0.000**	15	**0.988**	**0.977**	**0.059**	0.045 to 0.074	**0.0203**

χ^2^: chi-square; df: degrees of freedom; CFI: comparative fit index; TLI: Tucker–Lewis index; RMSEA: root mean square error of approximation; CI: confidence interval; SRMR: standardized root mean residual; F: factor; (C): crude model; (E): the model involves correlating residuals. Values in bold denote good fit.

**Table 3 ijerph-18-10142-t003:** Comparison of the normality of the Depression Anxiety Stress Scale 21 (DASS-21) and its subscales with that of the DASS-9/DASS-8 and their subscales.

	Samples	DASS-21	DASS-21 Depression	DASS-21 Anxiety	DASS-21 Stress	DASS-9	DASS-9 Depression	DASS-9 Anxiety ▲	DASS-9 Stress ▲	DASS-8	DASS-8 Depression
**Shapiro–Wilks’ W**	Sample 1	0.93	0.93	0.90	0.94	0.92	0.92	0.85	0.88	0.91	0.90
	Sample 2	0.83	0.83	0.72	0.84	0.82	0.86	0.67	0.75	0.80	0.82

**▲**: Both the anxiety depression subscales are the same on the DASS-9 and the DASS-8, all *p* values < 0.001.

**Table 4 ijerph-18-10142-t004:** Descriptive statistics, internal consistency, predictive validity, and discriminant validity of the Depression Anxiety Stress Scale 21 and its shortened versions.

DASS versions	Samples	*MD*	*Q1–Q3*	Coefficient Alpha	Alpha-If-Item-Deleted	Item-Total Correlations	Correlation with the DASS-21	U	W	z
**DASS-21**	Sample 1	21	6–39.8	0.959	0.956–0.959	0.364–0.784	--	51,198.5	542,734.5	−8.098
	Sample 2	7	2–17	0.939	0.933–0.940	0.172–0.696	--			
**Korean DASS-12**	Sample 1	12	5–21	**0.920**	0.906–0.919	**0.503–0.817**	**0.977**	50,933.5	542,469.5	−8.178
	Sample 2	5	1–10	**0.879**	0.856–0.881	**0.366–0.765**	**0.970**			
**Osman’s DASS-13**	Sample 1	12	4.3–24	0.928	0.918–0.927	0.512–0.794	0.984	50,478.5	542,014.5	−8.290
	Sample 2	5	1–10	0.890	0.872–0.893	0.331–0.766	0.971			
**Osman’s DASS-9**	Sample 1	9	3–16	0.886	0.868–0.884	0.491–0.708	0.967	50,506.0	542,042.0	−8.314
	Sample 2	3	1–7	0.829	0.789–0.831	0.335–0.708	0.921			
**Modified DASS-9**	Sample 1	10	3–19	**0.939**	0.928–0.936	0.683–0.830	**0.977**	51,697.0	543,233.0	−8.009
	Sample 2	3	1–8	**0.901**	0.883–0.905	0.471–0.757	**0.949**			
**DASS-8**	Sample 1	9	2–17	**0.936**	0.924–0.934	**0.688–0.826**	**0.972**	50,965.0	542,501.0	−8.229
	Sample 2	2	0–7	**0.905**	0.888–0.900	**0.625–0.756**	**0.929**			

MD: median, Q1: first quartile, Q2: third quartile, U: Mann Whitney U test, W: Wilcoxon test, all *p* values < 0.001. Bolded numbers represents wll-fitting models in this big table for a better readibility.

**Table 5 ijerph-18-10142-t005:** Internal consistency of subscales of the Depression Anxiety Stress Scale 21 and shortened versions in the samples.

Criteria	Samples	DASS-21			Korean DASS-12			DASS-13		Osman’s DASS-9			DASS-9			DASS-8
		Depression	Anxiety	Stress	Depression	Anxiety ▲	Stress	Depression	Stress	Depression	Anxiety	Stress	Depression	Anxiety △	Stress △	Depression
Coefficient alpha	Sample 1	0.902	0.872	0.908	0.850	0.772	0.852	0.898	0.824	0.833	0.731	0.695	0.869	0.888	0.835	0.854
	Sample 2	0.854	0.795	0.891	0.742	0.626	0.828	0.846	0.814	0.753	0.554	0.766	0.777	0.789	0.801	0.793
Range of corrected item-total correlations	Sample 1	0.607–0.797	0.496–0.748	0.577–0.820	0.653–0.741	0.497–0.641	0.579–0.766	0.665–0.793	0.599–0.780	0.604–0.708	0.484–0.611	0.533-	0.667–0.765	0.720–0.829	All 0.717	0.706–0.740
	Sample 2	0.468–0.707	0.349–0.667	0.574–0.788	0.416–0.606	0.350–0.459	0.561–0.768	0.443–0.707	0.614–0.744	0.404–0.631	0.319–0.400	0.623	0.443–0.659	0.580–0.696	All 0.668	0.595–0.673
Range of alpha if-item-deleted	Sample 1	0.877–0.898	0.840–0.873	0.884–0.910	0.788–0.826	0.680–0.755	0.779–0.856	0.869–0.889	0.695–0.835	0.770–0.815	0.580–0.722	--	0.820–0.845	0.801–0.895	-	0.782–0.814
	Sample 2	0.820–0.856	0.740–0.801	0.862–0.889	0.641–0.751	0.516–0.602	0.734–0.827	0.805–0.857	0.667–0.801	0.650–0.778	0.409–0.528	--	0.682–0.793	0.647–0.780	-	0.676–0.759
Correlation with the corresponding scale on the DASS-21	Sample 1	--	--	--	0.970	0.897	0.969	0.991	0.943	0.963	0.866	0.884	0.967	0.924	0.921	0.949
	Sample 2	--	--	--	0.947	0.828	0.955	0.979	0.899	0.906	0.799	0.832	0.958	0.837	0.863	0.899
Correlation with the DASS-21	Sample 1	0.952	0.935	0.958	0.918	0.796	0.926	0.943	0.895	0.911	0.769	0.830	0.918	0.902	0.881	0.904
	Sample 2	0.914	0.823	0.939	0.866	0.613	0.900	0.898	0.828	0.805	0.587	0.766	0.881	0.770	0.808	0.854

**▲**: Items on the anxiety subscale on the DASS-12 and the DASS-13 are the same, **△**: items on the anxiety and stress subscales on the modified DASS-9 and the DASS-8 are the same, all correlations are significant at the 0.01 level.

**Table 6 ijerph-18-10142-t006:** Descriptive statistics and discriminant validity of subscales of shortened versions of the Depression Anxiety Stress Scale 21.

DASS Subscales	Samples	Korean DASS-12					DASS-13					Osman’s DASS-9					Modified DASS-9				
		MD	Q1–Q3	U	W	z	MD	Q1–Q3	U	W	z	MD	Q1–Q3	U	W	z	MD	Q1–Q3	U	W	z
Depression	Sample 1	5	2–9	53,363.0	544,899.0	−7.659	7	2–13	51,796.0	543,332.0	−8.014	4	1–8	53,045.0	544,581.0	−7.826	5	2–9	54,378.5	545,914.5	−7.384
	Sample 2	2	0–4				3	0–5				1	0–3				2	0–4			
Anxiety	Sample 1	2	0–5	55,078.5	546,614.5	−7.724	2	0–5	55,078.5	546,614.5	−7.724	2	0–4	55,800.0	547,336.0	−7.609	3	0–6	54,555.0	546,091.0	−7.940
	Sample 2	0	0–2				0	0–2				0	0–1				0	0–2			
Stress	Sample 1	5	2–8	54,327.5	545,863.5	−7.424	4	1–6	56,008.5	547,544.5	−7.096	3	1–4	57,678.5	549,214.5	−6.780	2	0–4	56,053.5	547,586.5	−7.316
	Sample 2	2	0–5				1	0–3				1	0–3				0	0–2			

MD: median, Q1: first quartile, Q2: third quartile, U: Mann Whitney U test, W: Wilcoxon test, all *p* values < 0.001. For the depression subscale on the DASS-8, MD (Q1–Q2) = 4 (1–7, sample 1) and 1 (0–3, sample 2), U = 50,965.0, W = 542,501.0, z = −8.229.

## Data Availability

The dataset used to produce the current article [76] is publicly available at: https://data.mendeley.com/datasets/8k3vmfxpd3/draft?a=67415321-61f7-4920-bd2a-749b365ff6fb (access on 22 September 2021).

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
