# Peer review of "The Depression Anxiety Stress Scale 21: Development and Validation of the Depression Anxiety Stress Scale 8-Item in Psychiatric Patients and the General Public for Easier Mental Health Measurement in a Post COVID-19 World"

_ijerph, 2021, doi:10.3390/ijerph181910142_

Round 1

Reviewer 1 Report

The study examines the validity of an 8-item version of the DASS in Arabic
on an Egyptian population recruited online. Using a confirmatory factor analysis, the authors analyzed the validity of 5 short versions of the DASS compared to the original 21 item form. The patient sample (n = 168) was evaluated against that of the general population (n = 992). Multigroup CFA, Mann Whitney W test, Spear-man's correlation, and coefficient alpha were used to examine the shortened versions of the DASS-21 (DASS-13, DASS-12, DASS-9 (two versions), and DASS-8) for invariance across age and gender groups, discriminant validity, predictive validity, item coverage, and internal consistency, respectively. From the analyzes carried out by the authors it is concluded that the 8-item version has adequate psychometric properties compared to the 21-item version and that it can be used as a general screening tool in population studies, identifying a general condition of distress, while to evaluate the respective anxious and depressive components require further evaluation. I believe that the study is carried out with an adequate methodology and that the analyzes on the sample are carried out in a careful and precise way. The main problem with the study, which perhaps should be better highlighted within the limits of the work, is given by the methods of recruiting the sample, which were carried out online, without it being actually possible to clinically verify the diagnosis of the subpopulation defined as "Psychiatric" nor, consequently, that part of the sample that is defined as the "general population" is adequately assessed.

Author Response

Manuscript ID: ijerph-1375387

Response to the comments of Reviewer 1

We are very much grateful for the time and sincere help of the Reviewer. We have modified the whole manuscript taking into account the comments of the Reviewer, which are addressed line-by-line as shown below. Replies come underneath in red.

I believe that the study is carried out with an adequate methodology and that the analyzes on the sample are carried out in a careful and precise way. The main problem with the study, which perhaps should be better highlighted within the limits of the work, is given by the methods of recruiting the sample, which were carried out online, without it being actually possible to clinically verify the diagnosis of the subpopulation defined as "Psychiatric" nor, consequently, that part of the sample that is defined as the "general population" is adequately assessed.

Yes, we agree with reviewer. There is no guarantee that the respondents in the "general population" were mental-illness free since those respondents did not undergo a valid clinician-based diagnosis for mental disorders. In this sense, the discriminant validity of the DASS-21 and the shortened version may be questionable “quoted below”. We have noted this in the limitation section. Thank you so much.

“In the meantime, lack of screening of respondents in the community sample for mental disorders entails that some of those respondents may not be mental-illness free. This may cast doubt on the soundness of the comparisons of the psychometrics of measures, particularly discriminant validity, of the DASS-21 and its shortened versions between the two samples.”

We hope that the manuscript has been satisfactorily modified and that the current version will be suitable for publication.

Best regards,

Reviewer 2 Report

i write my comments in the file

Author Response

Manuscript ID: ijerph-1375387

Response to the comments of Reviewer 2

We are very much grateful for the time and sincere help of the Reviewer. We have modified the whole manuscript taking into account the comments of the Reviewer, which are addressed line-by-line as shown below. Replies come underneath in red.

  • Thank you for such a rich resource.

 - Regarding the sample, it is necessary to indicate a priori, in the methodology, the "requisites" for admission to the study of the sample with psychological problems, and the socio-demographic characteristics common to the two samples which allow, in the analysis, a comparison.

Yes, thank you for such an important comment. Actually, the participants were included in sample 1 or sample post hoc, after data collection was done. The only criteria used for this classification was participants’ subjective report of having a formally-diagnosed mental disorder. We have noted this in the first paragraph in the Methods “quoted below”. The socio-demographic characteristics common to the two samples were not used for classifying the samples. However, in each sample the socio-demographic characteristics were used for multigroup analysis without being compared between the two main samples. In this context, the performed comparisons within each sample seem to be rigorous.

“Among 1160 respondents, 168 reported having a preexisting mental disorder, which was diagnosed by a psychiatrist. Accordingly, respondents were classified based on the criteria of having or not having a psychiatric diagnosis into two samples: a psychiatric patient sample (sample 1) and a community sample (sample 2).”

- Limitations: The authors correctly report several limitations of the study. In my opinion, the most important limitation besides the sampling procedure concerns the numerical imbalance between the two samples. It is necessary that the authors also include this limitation.

Yes, we agree with reviewer. Sound comparisons should be based on randomly selected samples or at least selecting cross-matched controls. We have noted this in the limitation section. Thank you so much.

“In addition, the numerical imbalance between the two samples is another major limitation.”

We hope that the manuscript has been satisfactorily modified and that the current version will be suitable for publication.

Best regards,
